# Per-Pixel Classification is Not All You Need for Semantic Segmentation

**Bowen Cheng**[1,2*]   **Alexander G. Schwing**[2]   **Alexander Kirillov**[1]

[1]Facebook AI Research (FAIR)      [2]University of Illinois at Urbana-Champaign (UIUC)

## Abstract

Modern approaches typically formulate semantic segmentation as a *per-pixel classification* task, while instance-level segmentation is handled with an alternative *mask classification*. Our key insight: mask classification is sufficiently general to solve both semantic- and instance-level segmentation tasks in a unified manner using the exact same model, loss, and training procedure. Following this observation, we propose **MaskFormer**, a simple mask classification model which predicts a set of binary masks, each associated with a *single* global class label prediction. Overall, the proposed mask classification-based method simplifies the landscape of effective approaches to semantic and panoptic segmentation tasks and shows excellent empirical results. In particular, we observe that MaskFormer outperforms per-pixel classification baselines when the number of classes is large. Our mask classification-based method outperforms both current state-of-the-art semantic (55.6 mIoU on ADE20K) and panoptic segmentation (52.7 PQ on COCO) models.[1]

## 1   Introduction

The goal of semantic segmentation is to partition an image into regions with different semantic categories. Starting from Fully Convolutional Networks (FCNs) work of Long *et al*. [28], most *deep learning-based* semantic segmentation approaches formulate semantic segmentation as *per-pixel classification* (Figure 1 left), applying a classification loss to each output pixel [8, 46]. Per-pixel predictions in this formulation naturally partition an image into regions of different classes.

Mask classification is an alternative paradigm that disentangles the image partitioning and classification aspects of segmentation. Instead of classifying each pixel, mask classification-based methods predict a set of binary masks, each associated with a *single* class prediction (Figure 1 right). The more flexible mask classification dominates the field of instance-level segmentation. Both Mask R-CNN [19] and DETR [3] yield a single class prediction per segment for instance and panoptic segmentation. In contrast, per-pixel classification assumes a static number of outputs and cannot return a variable number of predicted regions/segments, which is required for instance-level tasks.

Our key observation: mask classification is sufficiently general to solve both semantic- and instance-level segmentation tasks. In fact, before FCN [28], the best performing semantic segmentation methods like O2P [4] and SDS [18] used a mask classification formulation. Given this perspective, a natural question emerges: *can a single mask classification model simplify the landscape of effective approaches to semantic- and instance-level segmentation tasks? And can such a mask classification model outperform existing per-pixel classification methods for semantic segmentation?*

To address both questions we propose a simple **MaskFormer** approach that seamlessly converts any existing per-pixel classification model into a mask classification. Using the set prediction mechanism proposed in DETR [3], MaskFormer employs a Transformer decoder [37] to compute a set of pairs,

---

*Work partly done during an internship at Facebook AI Research.

[1]Project page: https://bowenc0221.github.io/maskformer

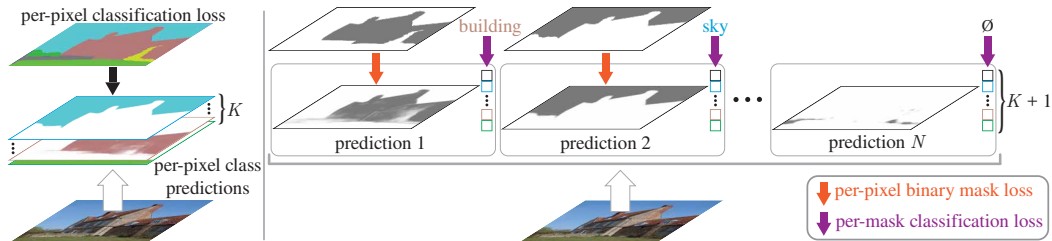

Figure 1: **Per-pixel classification *vs*. mask classification. (left)** Semantic segmentation with per-pixel classification applies the same classification loss to each location. **(right)** Mask classification predicts a set of binary masks and assigns a single class to each mask. Each prediction is supervised with a per-pixel binary mask loss and a classification loss. Matching between the set of predictions and ground truth segments can be done either via *bipartite matching* similarly to DETR [3] or by *fixed matching* via direct indexing if the number of predictions and classes match, *i.e.*, if $N = K$.

each consisting of a class prediction and a mask embedding vector. The mask embedding vector is used to get the binary mask prediction via a dot product with the per-pixel embedding obtained from an underlying fully-convolutional network. The new model solves both semantic- and instance-level segmentation tasks in a unified manner: no changes to the model, losses, and training procedure are required. Specifically, for semantic and panoptic segmentation tasks alike, MaskFormer is supervised with the same per-pixel binary mask loss and a single classification loss per mask. Finally, we design a simple inference strategy to blend MaskFormer outputs into a task-dependent prediction format.

We evaluate MaskFormer on five semantic segmentation datasets with various numbers of categories: Cityscapes [13] (19 classes), Mapillary Vistas [31] (65 classes), ADE20K [49] (150 classes), COCO-Stuff-10K [2] (171 classes), and ADE20K-Full [49] (847 classes). While MaskFormer performs on par with per-pixel classification models for Cityscapes, which has a few diverse classes, the new model demonstrates superior performance for datasets with larger vocabulary. We hypothesize that a single class prediction per mask models fine-grained recognition better than per-pixel class predictions. MaskFormer achieves the new state-of-the-art on ADE20K (**55.6 mIoU**) with Swin-Transformer [27] backbone, outperforming a per-pixel classification model [27] with the same backbone by 2.1 mIoU, while being more efficient (10% reduction in parameters and 40% reduction in FLOPs).

Finally, we study MaskFormer's ability to solve instance-level tasks using two panoptic segmentation datasets: COCO [26, 22] and ADE20K [49]. MaskFormer outperforms a more complex DETR model [3] with the same backbone and the same post-processing. Moreover, MaskFormer achieves the new state-of-the-art on COCO (**52.7 PQ**), outperforming prior state-of-the-art [38] by 1.6 PQ. Our experiments highlight MaskFormer's ability to unify instance- and semantic-level segmentation.

## 2  Related Works

Both per-pixel classification and mask classification have been extensively studied for semantic segmentation. In early work, Konishi and Yuille [23] apply per-pixel Bayesian classifiers based on local image statistics. Then, inspired by early works on non-semantic groupings [11, 33], mask classification-based methods became popular demonstrating the best performance in PASCAL VOC challenges [16]. Methods like O2P [4] and CFM [14] have achieved state-of-the-art results by classifying mask proposals [5, 36, 1]. In 2015, FCN [28] extended the idea of per-pixel classification to deep nets, significantly outperforming all prior methods on mIoU (a per-pixel evaluation metric which particularly suits the per-pixel classification formulation of segmentation).

**Per-pixel classification** became the dominant way for *deep-net-based* semantic segmentation since the seminal work of Fully Convolutional Networks (FCNs) [28]. Modern semantic segmentation models focus on aggregating long-range context in the final feature map: ASPP [6, 7] uses atrous convolutions with different atrous rates; PPM [46] uses pooling operators with different kernel sizes; DANet [17], OCNet [45], and CCNet [21] use different variants of non-local blocks [39]. Recently, SETR [47] and Segmenter [34] replace traditional convolutional backbones with Vision Transformers (ViT) [15] that capture long-range context starting from the very first layer. However, these concurrent Transformer-based [37] semantic segmentation approaches still use a per-pixel classification

formulation. Note, that our MaskFormer module can convert any per-pixel classification model to the mask classification setting, allowing seamless adoption of advances in per-pixel classification.

**Mask classification** is commonly used for instance-level segmentation tasks [18, 22]. These tasks require a dynamic number of predictions, making application of per-pixel classification challenging as it assumes a static number of outputs. Omnipresent Mask R-CNN [19] uses a global classifier to classify mask proposals for instance segmentation. DETR [3] further incorporates a Transformer [37] design to handle thing and stuff segmentation simultaneously for panoptic segmentation [22]. However, these mask classification methods require predictions of bounding boxes, which may limit their usage in semantic segmentation. The recently proposed Max-DeepLab [38] removes the dependence on box predictions for panoptic segmentation with conditional convolutions [35, 40]. However, in addition to the main mask classification losses it requires multiple auxiliary losses (*i.e.*, instance discrimination loss, mask-ID cross entropy loss, and the standard per-pixel classification loss).

## 3 From Per-Pixel to Mask Classification

In this section, we first describe how semantic segmentation can be formulated as either a per-pixel classification or a mask classification problem. Then, we introduce our instantiation of the mask classification model with the help of a Transformer decoder [37]. Finally, we describe simple inference strategies to transform mask classification outputs into task-dependent prediction formats.

### 3.1 Per-pixel classification formulation

For per-pixel classification, a segmentation model aims to predict the probability distribution over all possible $K$ categories for every pixel of an $H \times W$ image: $y = \{p_i | p_i \in \Delta^K\}_{i=1}^{H \cdot W}$. Here $\Delta^K$ is the $K$-dimensional probability simplex. Training a per-pixel classification model is straight-forward: given ground truth category labels $y^{\text{gt}} = \{y_i^{\text{gt}} | y_i^{\text{gt}} \in \{1, \ldots, K\}\}_{i=1}^{H \cdot W}$ for every pixel, a per-pixel cross-entropy (negative log-likelihood) loss is usually applied, *i.e.*, $\mathcal{L}_{\text{pixel-cls}}(y, y^{\text{gt}}) = \sum_{i=1}^{H \cdot W} - \log p_i(y_i^{\text{gt}})$.

### 3.2 Mask classification formulation

Mask classification splits the segmentation task into 1) partitioning/grouping the image into $N$ regions ($N$ does not need to equal $K$), represented with binary masks $\{m_i | m_i \in [0, 1]^{H \times W}\}_{i=1}^N$; and 2) associating each region as a whole with some distribution over $K$ categories. To jointly group and classify a segment, *i.e.*, to perform mask classification, we define the desired output $z$ as a set of $N$ probability-mask pairs, *i.e.*, $z = \{(p_i, m_i)\}_{i=1}^N$. In contrast to per-pixel class probability prediction, for mask classification the probability distribution $p_i \in \Delta^{K+1}$ contains an auxiliary "no object" label ($\varnothing$) in addition to the $K$ category labels. The $\varnothing$ label is predicted for masks that do not correspond to any of the $K$ categories. Note, mask classification allows multiple mask predictions with the same associated class, making it applicable to both semantic- and instance-level segmentation tasks.

To train a mask classification model, a matching $\sigma$ between the set of predictions $z$ and the set of $N^{\text{gt}}$ ground truth segments $z^{\text{gt}} = \{(c_i^{\text{gt}}, m_i^{\text{gt}}) | c_i^{\text{gt}} \in \{1, \ldots, K\}, m_i^{\text{gt}} \in \{0, 1\}^{H \times W}\}_{i=1}^{N^{\text{gt}}}$ is required.[2] Here $c_i^{\text{gt}}$ is the ground truth class of the $i^{\text{th}}$ ground truth segment. Since the size of prediction set $|z| = N$ and ground truth set $|z^{\text{gt}}| = N^{\text{gt}}$ generally differ, we assume $N \geq N^{\text{gt}}$ and pad the set of ground truth labels with "no object" tokens $\varnothing$ to allow one-to-one matching.

For semantic segmentation, a trivial *fixed matching* is possible if the number of predictions $N$ matches the number of category labels $K$. In this case, the $i^{\text{th}}$ prediction is matched to a ground truth region with class label $i$ and to $\varnothing$ if a region with class label $i$ is not present in the ground truth. In our experiments, we found that a *bipartite matching*-based assignment demonstrates better results than the fixed matching. Unlike DETR [3] that uses bounding boxes to compute the assignment costs between prediction $z_i$ and ground truth $z_j^{\text{gt}}$ for the matching problem, we directly use class and mask predictions, *i.e.*, $-p_i(c_j^{\text{gt}}) + \mathcal{L}_{\text{mask}}(m_i, m_j^{\text{gt}})$, where $\mathcal{L}_{\text{mask}}$ is a binary mask loss.

To train model parameters, given a matching, the main mask classification loss $\mathcal{L}_{\text{mask-cls}}$ is composed of a cross-entropy classification loss and a binary mask loss $\mathcal{L}_{\text{mask}}$ for each predicted segment:

$$\mathcal{L}_{\text{mask-cls}}(z, z^{\text{gt}}) = \sum_{j=1}^N \left[ - \log p_{\sigma(j)}(c_j^{\text{gt}}) + \mathbb{1}_{c_j^{\text{gt}} \neq \varnothing} \mathcal{L}_{\text{mask}}(m_{\sigma(j)}, m_j^{\text{gt}}) \right]. \tag{1}$$

---

[2]Different mask classification methods utilize various matching rules. For instance, Mask R-CNN [19] uses a heuristic procedure based on anchor boxes and DETR [3] optimizes a bipartite matching between $z$ and $z^{\text{gt}}$.

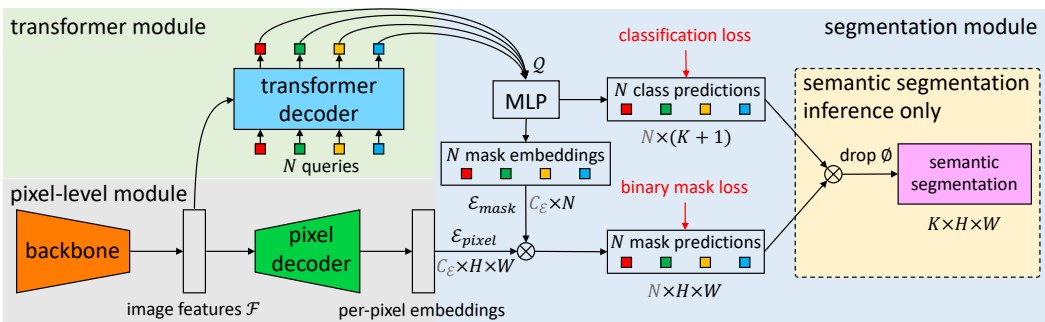

Figure 2: **MaskFormer overview.** We use a backbone to extract image features $\mathcal{F}$. A pixel decoder gradually upsamples image features to extract per-pixel embeddings $\mathcal{E}_{\text{pixel}}$. A transformer decoder attends to image features and produces $N$ per-segment embeddings $\mathcal{Q}$. The embeddings independently generate $N$ class predictions with $N$ corresponding mask embeddings $\mathcal{E}_{\text{mask}}$. Then, the model predicts $N$ possibly overlapping binary mask predictions via a dot product between pixel embeddings $\mathcal{E}_{\text{pixel}}$ and mask embeddings $\mathcal{E}_{\text{mask}}$ followed by a sigmoid activation. For semantic segmentation task we can get the final prediction by combining $N$ binary masks with their class predictions using a simple matrix multiplication (see Section 3.4). Note, the dimensions for multiplication $\otimes$ are shown in gray.

Note, that most existing mask classification models use auxiliary losses (*e.g.*, a bounding box loss [19, 3] or an instance discrimination loss [38]) in addition to $\mathcal{L}_{\text{mask-cls}}$. In the next section we present a simple mask classification model that allows end-to-end training with $\mathcal{L}_{\text{mask-cls}}$ alone.

### 3.3 MaskFormer

We now introduce MaskFormer, the new mask classification model, which computes $N$ probability-mask pairs $z = \{(p_i, m_i)\}_{i=1}^{N}$. The model contains three modules (see Fig. 2): 1) a pixel-level module that extracts per-pixel embeddings used to generate binary mask predictions; 2) a transformer module, where a stack of Transformer decoder layers [37] computes $N$ per-segment embeddings; and 3) a segmentation module, which generates predictions $\{(p_i, m_i)\}_{i=1}^{N}$ from these embeddings. During inference, discussed in Sec. 3.4, $p_i$ and $m_i$ are assembled into the final prediction.

**Pixel-level module** takes an image of size $H \times W$ as input. A backbone generates a (typically) low-resolution image feature map $\mathcal{F} \in \mathbb{R}^{C_{\mathcal{F}} \times \frac{H}{S} \times \frac{W}{S}}$, where $C_{\mathcal{F}}$ is the number of channels and $S$ is the stride of the feature map ($C_{\mathcal{F}}$ depends on the specific backbone and we use $S = 32$ in this work). Then, a pixel decoder gradually upsamples the features to generate per-pixel embeddings $\mathcal{E}_{\text{pixel}} \in \mathbb{R}^{C_{\mathcal{E}} \times H \times W}$, where $C_{\mathcal{E}}$ is the embedding dimension. Note, that any per-pixel classification-based segmentation model fits the pixel-level module design including recent Transformer-based models [34, 47, 27]. MaskFormer seamlessly converts such a model to mask classification.

**Transformer module** uses the standard Transformer decoder [37] to compute from image features $\mathcal{F}$ and $N$ learnable positional embeddings (*i.e.*, queries) its output, *i.e.*, $N$ per-segment embeddings $\mathcal{Q} \in \mathbb{R}^{C_{\mathcal{Q}} \times N}$ of dimension $C_{\mathcal{Q}}$ that encode global information about each segment MaskFormer predicts. Similarly to [3], the decoder yields all predictions in parallel.

**Segmentation module** applies a linear classifier, followed by a softmax activation, on top of the per-segment embeddings $\mathcal{Q}$ to yield class probability predictions $\{p_i \in \Delta^{K+1}\}_{i=1}^{N}$ for each segment. Note, that the classifier predicts an additional "no object" category ($\varnothing$) in case the embedding does not correspond to any region. For mask prediction, a Multi-Layer Perceptron (MLP) with 2 hidden layers converts the per-segment embeddings $\mathcal{Q}$ to $N$ mask embeddings $\mathcal{E}_{\text{mask}} \in \mathbb{R}^{C_{\mathcal{E}} \times N}$ of dimension $C_{\mathcal{E}}$. Finally, we obtain each binary mask prediction $m_i \in [0, 1]^{H \times W}$ via a dot product between the $i^{\text{th}}$ mask embedding and per-pixel embeddings $\mathcal{E}_{\text{pixel}}$ computed by the pixel-level module. The dot product is followed by a sigmoid activation, *i.e.*, $m_i[h, w] = \text{sigmoid}(\mathcal{E}_{\text{mask}}[:, i]^{\text{T}} \cdot \mathcal{E}_{\text{pixel}}[:, h, w])$.

Note, we empirically find it is beneficial to *not* enforce mask predictions to be mutually exclusive to each other by using a softmax activation. During training, the $\mathcal{L}_{\text{mask-cls}}$ loss combines a cross entropy classification loss and a binary mask loss $\mathcal{L}_{\text{mask}}$ for each predicted segment. For simplicity we use the same $\mathcal{L}_{\text{mask}}$ as DETR [3], *i.e.*, a linear combination of a focal loss [25] and a dice loss [30] multiplied by hyper-parameters $\lambda_{\text{focal}}$ and $\lambda_{\text{dice}}$ respectively.

### 3.4 Mask-classification inference

First, we present a simple *general inference* procedure that converts mask classification outputs $\{(p_i, m_i)\}_{i=1}^{N}$ to either panoptic or semantic segmentation output formats. Then, we describe a *semantic inference* procedure specifically designed for semantic segmentation. We note, that the specific choice of inference strategy largely depends on the evaluation metric rather than the task.

**General inference** partitions an image into segments by assigning each pixel $[h, w]$ to one of the $N$ predicted probability-mask pairs via $\arg\max_{i:c_i \neq \varnothing} p_i(c_i) \cdot m_i[h, w]$. Here $c_i$ is the most likely class label $c_i = \arg\max_{c \in \{1,...,K,\varnothing\}} p_i(c)$ for each probability-mask pair $i$. Intuitively, this procedure assigns a pixel at location $[h, w]$ to probability-mask pair $i$ only if both the *most likely* class probability $p_i(c_i)$ and the mask prediction probability $m_i[h, w]$ are high. Pixels assigned to the same probability-mask pair $i$ form a segment where each pixel is labelled with $c_i$. For semantic segmentation, segments sharing the same category label are merged; whereas for instance-level segmentation tasks, the index $i$ of the probability-mask pair helps to distinguish different instances of the same class. Finally, to reduce false positive rates in panoptic segmentation we follow previous inference strategies [3, 22]. Specifically, we filter out low-confidence predictions prior to inference and remove predicted segments that have large parts of their binary masks ($m_i > 0.5$) occluded by other predictions.

**Semantic inference** is designed specifically for semantic segmentation and is done via a simple matrix multiplication. We empirically find that marginalization over probability-mask pairs, *i.e.*, $\arg\max_{c \in \{1,...,K\}} \sum_{i=1}^{N} p_i(c) \cdot m_i[h, w]$, yields better results than the hard assignment of each pixel to a probability-mask pair $i$ used in the general inference strategy. The argmax does not include the "no object" category ($\varnothing$) as standard semantic segmentation requires each output pixel to take a label. Note, this strategy returns a per-pixel class probability $\sum_{i=1}^{N} p_i(c) \cdot m_i[h, w]$. However, we observe that directly maximizing per-pixel class likelihood leads to poor performance. We hypothesize, that gradients are evenly distributed to every query, which complicates training.

## 4 Experiments

We demonstrate that MaskFormer seamlessly unifies semantic- and instance-level segmentation tasks by showing state-of-the-art results on both semantic segmentation and panoptic segmentation datasets. Then, we ablate the MaskFormer design confirming that observed improvements in semantic segmentation indeed stem from the shift from per-pixel classification to mask classification.

**Datasets.** We study MaskFormer using four widely used semantic segmentation datasets: ADE20K [49] (150 classes) from the SceneParse150 challenge [48], COCO-Stuff-10K [2] (171 classes), Cityscapes [13] (19 classes), and Mapillary Vistas [31] (65 classes). In addition, we use the ADE20K-Full [49] dataset annotated in an open vocabulary setting (we keep 874 classes that are present in both train and validation sets). For panotic segmenation evaluation we use COCO [26, 2, 22] (80 "things" and 53 "stuff" categories) and ADE20K-Panoptic [49, 22] (100 "things" and 50 "stuff" categories). Please see the appendix for detailed descriptions of all used datasets.

**Evaluation metrics.** For *semantic segmentation* the standard metric is **mIoU** (mean Intersection-over-Union) [16], a per-pixel metric that directly corresponds to the per-pixel classification formulation. To better illustrate the difference between segmentation approaches, in our ablations we supplement mIoU with **PQ$^{\text{St}}$** (PQ stuff) [22], a per-region metric that treats all classes as "stuff" and evaluates each segment equally, irrespective of its size. We report the median of 3 runs for all datasets, except for Cityscapes where we report the median of 5 runs. For *panoptic segmentation*, we use the standard **PQ** (panoptic quality) metric [22] and report single run results due to prohibitive training costs.

**Baseline models.** On the right we sketch the used per-pixel classification baselines. The **PerPixelBaseline** uses the pixel-level module of MaskFormer and directly outputs per-pixel class scores. For a fair comparison, we design **PerPixelBaseline+** which adds the transformer module and mask embedding MLP to the PerPixelBaseline. Thus, PerPixelBaseline+ and MaskFormer differ only in the formulation: per-pixel *vs.* mask classification. Note that these baselines are for ablation and we compare MaskFormer with state-of-the-art per-pixel classification models as well.

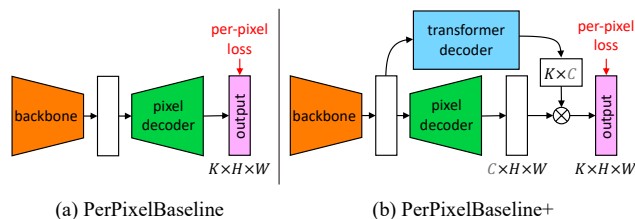

(a) PerPixelBaseline      (b) PerPixelBaseline+

## 4.1 Implementation details

**Backbone.** MaskFormer is compatible with any backbone architecture. In our work we use the standard convolution-based ResNet [20] backbones (R50 and R101 with 50 and 101 layers respectively) and recently proposed Transformer-based Swin-Transformer [27] backbones. In addition, we use the R101c model [6] which replaces the first $7 \times 7$ convolution layer of R101 with 3 consecutive $3 \times 3$ convolutions and which is popular in the semantic segmentation community [46, 7, 8, 21, 44, 10].

**Pixel decoder.** The pixel decoder in Figure 2 can be implemented using any semantic segmentation decoder (*e.g.*, [8–10]). Many per-pixel classification methods use modules like ASPP [6] or PSP [46] to collect and distribute context across locations. The Transformer module attends to all image features, collecting global information to generate class predictions. This setup reduces the need of the per-pixel module for heavy context aggregation. Therefore, for MaskFormer, we design a light-weight pixel decoder based on the popular FPN [24] architecture.

Following FPN, we $2\times$ upsample the low-resolution feature map in the decoder and sum it with the projected feature map of corresponding resolution from the backbone; Projection is done to match channel dimensions of the feature maps with a $1 \times 1$ convolution layer followed by GroupNorm (GN) [41]. Next, we fuse the summed features with an additional $3 \times 3$ convolution layer followed by GN and ReLU activation. We repeat this process starting with the stride 32 feature map until we obtain a final feature map of stride 4. Finally, we apply a single $1 \times 1$ convolution layer to get the per-pixel embeddings. All feature maps in the pixel decoder have a dimension of 256 channels.

**Transformer decoder.** We use the same Transformer decoder design as DETR [3]. The $N$ query embeddings are initialized as zero vectors, and we associate each query with a learnable positional encoding. We use 6 Transformer decoder layers with 100 queries by default, and, following DETR, we apply the same loss after each decoder. In our experiments we observe that MaskFormer is competitive for semantic segmentation with a single decoder layer too, whereas for instance-level segmentation multiple layers are necessary to remove duplicates from the final predictions.

**Segmentation module.** The multi-layer perceptron (MLP) in Figure 2 has 2 hidden layers of 256 channels to predict the mask embeddings $\mathcal{E}_{\text{mask}}$, analogously to the box head in DETR. Both per-pixel $\mathcal{E}_{\text{pixel}}$ and mask $\mathcal{E}_{\text{mask}}$ embeddings have 256 channels.

**Loss weights.** We use focal loss [25] and dice loss [30] for our mask loss: $\mathcal{L}_{\text{mask}}(m, m^{\text{gt}}) = \lambda_{\text{focal}}\mathcal{L}_{\text{focal}}(m, m^{\text{gt}}) + \lambda_{\text{dice}}\mathcal{L}_{\text{dice}}(m, m^{\text{gt}})$, and set the hyper-parameters to $\lambda_{\text{focal}} = 20.0$ and $\lambda_{\text{dice}} = 1.0$. Following DETR [3], the weight for the "no object" ($\varnothing$) in the classification loss is set to 0.1.

## 4.2 Training settings

**Semantic segmentation.** We use Detectron2 [42] and follow the commonly used training settings for each dataset. More specifically, we use AdamW [29] and the *poly* [6] learning rate schedule with an initial learning rate of $10^{-4}$ and a weight decay of $10^{-4}$ for ResNet [20] backbones, and an initial learning rate of $6 \cdot 10^{-5}$ and a weight decay of $10^{-2}$ for Swin-Transformer [27] backbones. Backbones are pre-trained on ImageNet-1K [32] if not stated otherwise. A learning rate multiplier of 0.1 is applied to CNN backbones and 1.0 is applied to Transformer backbones. The standard random scale jittering between 0.5 and 2.0, random horizontal flipping, random cropping as well as random color jittering are used as data augmentation [12]. For the ADE20K dataset, if not stated otherwise, we use a crop size of $512 \times 512$, a batch size of 16 and train all models for 160k iterations. For the ADE20K-Full dataset, we use the same setting as ADE20K except that we train all models for 200k iterations. For the COCO-Stuff-10k dataset, we use a crop size of $640 \times 640$, a batch size of 32 and train all models for 60k iterations. All models are trained with 8 V100 GPUs. We report both performance of single scale (s.s.) inference and multi-scale (m.s.) inference with horizontal flip and scales of 0.5, 0.75, 1.0, 1.25, 1.5, 1.75. See appendix for Cityscapes and Mapillary Vistas settings.

**Panoptic segmentation.** We follow exactly the same architecture, loss, and training procedure as we use for semantic segmentation. The only difference is supervision: *i.e.*, category region masks in semantic segmentation *vs*. object instance masks in panoptic segmentation. We strictly follow the DETR [3] setting to train our model on the COCO panoptic segmentation dataset [22] for a fair comparison. On the ADE20K panoptic segmentation dataset, we follow the semantic segmentation setting but train for longer (720k iterations) and use a larger crop size ($640 \times 640$). COCO models are trained using 64 V100 GPUs and ADE20K experiments are trained with 8 V100 GPUs. We use

Table 1: **Semantic segmentation on ADE20K** `val` **with 150 categories.** Mask classification-based MaskFormer outperforms the best per-pixel classification approaches while using fewer parameters and less computation. We report both single-scale (s.s.) and multi-scale (m.s.) inference results with ±*std*. FLOPs are computed for the given crop size. Frames-per-second (fps) is measured on a V100 GPU with a batch size of 1.[3] Backbones pre-trained on ImageNet-22K are marked with [†].

| | method | backbone | crop size | mIoU (s.s.) | mIoU (m.s.) | #params. | FLOPs | fps |
|---|---|---|---|---|---|---|---|---|
| CNN backbones | OCRNet [44] | R101c | $520 \times 520$ | - | 45.3 | - | - | - |
| | DeepLabV3+ [8] | R50c | $512 \times 512$ | 44.0 | 44.9 | 44M | 177G | 21.0 |
| | | R101c | $512 \times 512$ | 45.5 | 46.4 | 63M | 255G | 14.2 |
| | **MaskFormer** (ours) | R50 | $512 \times 512$ | 44.5 ±0.5 | 46.7 ±0.6 | 41M | 53G | 24.5 |
| | | R101 | $512 \times 512$ | 45.5 ±0.5 | 47.2 ±0.2 | 60M | 73G | 19.5 |
| | | R101c | $512 \times 512$ | **46.0** ±0.1 | **48.1** ±0.2 | 60M | 80G | 19.0 |
| Transformer backbones | SETR [47] | ViT-L[†] | $512 \times 512$ | - | 50.3 | 308M | - | - |
| | Swin-UperNet [27, 43] | Swin-T | $512 \times 512$ | - | 46.1 | 60M | 236G | 18.5 |
| | | Swin-S | $512 \times 512$ | - | 49.3 | 81M | 259G | 15.2 |
| | | Swin-B[†] | $640 \times 640$ | - | 51.6 | 121M | 471G | 8.7 |
| | | Swin-L[†] | $640 \times 640$ | - | 53.5 | 234M | 647G | 6.2 |
| | **MaskFormer** (ours) | Swin-T | $512 \times 512$ | 46.7 ±0.7 | 48.8 ±0.6 | 42M | 55G | 22.1 |
| | | Swin-S | $512 \times 512$ | 49.8 ±0.4 | 51.0 ±0.4 | 63M | 79G | 19.6 |
| | | Swin-B | $640 \times 640$ | 51.1 ±0.2 | 52.3 ±0.4 | 102M | 195G | 12.6 |
| | | Swin-B[†] | $640 \times 640$ | 52.7 ±0.4 | 53.9 ±0.2 | 102M | 195G | 12.6 |
| | | Swin-L[†] | $640 \times 640$ | **54.1** ±0.2 | **55.6** ±0.1 | 212M | 375G | 7.9 |

Table 2: **MaskFormer** *vs.* **per-pixel classification baselines on 4 semantic segmentation datasets.** MaskFormer improvement is larger when the number of classes is larger. We use a ResNet-50 backbone and report single scale mIoU and PQ$^{St}$ for ADE20K, COCO-Stuff and ADE20K-Full, whereas for higher-resolution Cityscapes we use a deeper ResNet-101 backbone following [7, 8].

| | Cityscapes (19 classes) | | ADE20K (150 classes) | | COCO-Stuff (171 classes) | | ADE20K-Full (847 classes) | |
|---|---|---|---|---|---|---|---|---|
| | mIoU | PQ$^{St}$ | mIoU | PQ$^{St}$ | mIoU | PQ$^{St}$ | mIoU | PQ$^{St}$ |
| PerPixelBaseline | 77.4 | 58.9 | 39.2 | 21.6 | 32.4 | 15.5 | 12.4 | 5.8 |
| PerPixelBaseline+ | **78.5** | 60.2 | 41.9 | 28.3 | 34.2 | 24.6 | 13.9 | 9.0 |
| **MaskFormer** (ours) | **78.5** (+0.0) | **63.1** (+2.9) | **44.5** (+2.6) | **33.4** (+5.1) | **37.1** (+2.9) | **28.9** (+4.3) | **17.4** (+3.5) | **11.9** (+2.9) |

the general inference (Section 3.4) with the following parameters: we filter out masks with class confidence below 0.8 and set masks whose contribution to the final panoptic segmentation is less than 80% of its mask area to VOID. We report performance of single scale inference.

## 4.3 Main results

**Semantic segmentation.** In Table 1, we compare MaskFormer with state-of-the-art per-pixel classification models for semantic segmentation on the ADE20K `val` set. With the same standard CNN backbones (*e.g.*, ResNet [20]), MaskFormer outperforms DeepLabV3+ [8] by 1.7 mIoU. MaskFormer is also compatible with recent Vision Transformer [15] backbones (*e.g.*, the Swin Transformer [27]), achieving a new state-of-the-art of 55.6 mIoU, which is 2.1 mIoU better than the prior state-of-the-art [27]. Observe that MaskFormer outperforms the best per-pixel classification-based models while having fewer parameters and faster inference time. This result suggests that the mask classification formulation has significant potential for semantic segmentation. See appendix for results on `test` set.

Beyond ADE20K, we further compare MaskFormer with our baselines on COCO-Stuff-10K, ADE20K-Full as well as Cityscapes in Table 2 and we refer to the appendix for comparison with state-of-the-art methods on these datasets. The improvement of MaskFormer over PerPixelBaseline+ is larger when the number of classes is larger: For Cityscapes, which has only 19 categories, MaskFormer performs similarly well as PerPixelBaseline+; While for ADE20K-Full, which has 847 classes, MaskFormer outperforms PerPixelBaseline+ by 3.5 mIoU.

Although MaskFormer shows no improvement in mIoU for Cityscapes, the PQ$^{St}$ metric increases by 2.9 PQ$^{St}$. We find MaskFormer performs better in terms of recognition quality (RQ$^{St}$) while lagging in per-pixel segmentation quality (SQ$^{St}$) (we refer to the appendix for detailed numbers). This observation suggests that on datasets where class recognition is relatively easy to solve, the main challenge for mask classification-based approaches is pixel-level accuracy (*i.e.*, mask quality).

---

[3]It isn't recommended to compare fps from different papers: speed is measured in different environments. DeepLabV3+ fps are from MMSegmentation [12], and Swin-UperNet fps are from the original paper [27].

Table 3: **Panoptic segmentation on COCO panoptic** `val` **with 133 categories.** MaskFormer seamlessly unifies semantic- and instance-level segmentation without modifying the model architecture or loss. Our model, which achieves better results, can be regarded as a box-free simplification of DETR [3]. The major improvement comes from "stuff" classes ($PQ^{St}$) which are ambiguous to represent with bounding boxes. For MaskFormer (DETR) we use the exact same post-processing as DETR. Note, that in this setting MaskFormer performance is still better than DETR (+2.2 PQ). Our model also outperforms recently proposed Max-DeepLab [38] without the need of sophisticated auxiliary losses, while being more efficient. FLOPs are computed as the average FLOPs over 100 validation images (COCO images have varying sizes). Frames-per-second (fps) is measured on a V100 GPU with a batch size of 1 by taking the average runtime on the entire `val` set *including post-processing time*. Backbones pre-trained on ImageNet-22K are marked with $^\dagger$.

| | method | backbone | PQ | $PQ^{Th}$ | $PQ^{St}$ | SQ | RQ | #params. | FLOPs | fps |
|---|---|---|---|---|---|---|---|---|---|---|
| CNN backbones | DETR [3] | R50 + 6 Enc | 43.4 | 48.2 | 36.3 | 79.3 | 53.8 | - | - | - |
| | MaskFormer (DETR) | R50 + 6 Enc | 45.6 | 50.0 (+1.8) | 39.0 (+2.7) | 80.2 | 55.8 | - | - | - |
| | **MaskFormer** (ours) | R50 + 6 Enc | **46.5** | **51.0** (+2.8) | **39.8** (+3.5) | **80.4** | **56.8** | 45M | 181G | 17.6 |
| | DETR [3] | R101 + 6 Enc | 45.1 | 50.5 | 37.0 | 79.9 | 55.5 | - | - | - |
| | **MaskFormer** (ours) | R101 + 6 Enc | **47.6** | **52.5** (+2.0) | **40.3** (+3.3) | **80.7** | **58.0** | 64M | 248G | 14.0 |
| Transformer backbones | Max-DeepLab [38] | Max-S | 48.4 | 53.0 | 41.5 | - | - | 62M | 324G | 7.6 |
| | | Max-L | 51.1 | 57.0 | 42.2 | - | - | 451M | 3692G | - |
| | **MaskFormer** (ours) | Swin-T | 47.7 | 51.7 | 41.7 | 80.4 | 58.3 | 42M | 179G | 17.0 |
| | | Swin-S | 49.7 | 54.4 | 42.6 | 80.9 | 60.4 | 63M | 259G | 12.4 |
| | | Swin-B | 51.1 | 56.3 | 43.2 | 81.4 | 61.8 | 102M | 411G | 8.4 |
| | | Swin-B$^\dagger$ | 51.8 | 56.9 | **44.1** | 81.4 | 62.6 | 102M | 411G | 8.4 |
| | | Swin-L$^\dagger$ | **52.7** | **58.5** | 44.0 | **81.8** | **63.5** | 212M | 792G | 5.2 |

**Panoptic segmentation.** In Table 3, we compare the same exact MaskFormer model with DETR [3] on the COCO panoptic `val` set. To match the standard DETR design, we add 6 additional Transformer encoder layers after the CNN backbone. Unlike DETR, our model does not predict bounding boxes but instead predicts masks directly. MaskFormer achieves better results while being simpler than DETR. To disentangle the improvements from the model itself and our post-processing inference strategy we run our model following DETR post-processing (MaskFormer (DETR)) and observe that this setup outperforms DETR by 2.2 PQ. Overall, we observe a larger improvement in $PQ^{St}$ compared to $PQ^{Th}$. This suggests that detecting "stuff" with bounding boxes is suboptimal, and therefore, box-based segmentation models (*e.g.*, Mask R-CNN [19]) do not suit semantic segmentation. MaskFormer also outperforms recently proposed Max-DeepLab [38] without the need of special network design as well as sophisticated auxiliary losses (*i.e.*, instance discrimination loss, mask-ID cross entropy loss, and per-pixel classification loss in [38]). *MaskFormer, for the first time, unifies semantic- and instance-level segmentation with the exact same model, loss, and training pipeline.*

We further evaluate our model on the panoptic segmentation version of the ADE20K dataset. Our model also achieves state-of-the-art performance. We refer to the appendix for detailed results.

### 4.4 Ablation studies

We perform a series of ablation studies of MaskFormer using a single ResNet-50 backbone [20].

**Per-pixel *vs*. mask classification.** In Table 4, we verify that the gains demonstrated by MaskFromer come from shifting the paradigm to mask classification. We start by comparing PerPixelBaseline+ and MaskFormer. The models are very similar and there are only 3 differences: 1) per-pixel *vs*. mask classification used by the models, 2) MaskFormer uses bipartite matching, and 3) the new model uses a combination of focal and dice losses as a mask loss, whereas PerPixelBaseline+ utilizes per-pixel cross entropy loss. First, we rule out the influence of loss differences by training PerPixelBaseline+ with exactly the same losses and observing no improvement. Next, in Table 4a, we compare PerPixelBaseline+ with MaskFormer trained using a fixed matching (MaskFormer-fixed), *i.e.*, $N = K$ and assignment done based on category label indices identically to the per-pixel classification setup. We observe that MaskFormer-fixed is 1.8 mIoU better than the baseline, suggesting that shifting from per-pixel classification to mask classification is indeed the main reason for the gains of MaskFormer. In Table 4b, we further compare MaskFormer-fixed with MaskFormer trained with bipartite matching (MaskFormer-bipartite) and find bipartite matching is not only more flexible (allowing to predict less masks than the total number of categories) but also produces better results.

Table 4: **Per-pixel _vs_. mask classification for semantic segmentation.** All models use 150 queries for a fair comparison. We evaluate the models on ADE20K val with 150 categories. 4a: PerPixel-Baseline+ and MaskFormer-fixed use similar fixed matching (_i.e._, matching by category index), this result confirms that the shift from per-pixel to mask classification is the key. 4b: bipartite matching is not only more flexible (can make less prediction than total class count) but also gives better results.

(a) Per-pixel _vs_. mask classification.

| | mIoU | PQ$^{St}$ |
|---|---|---|
| PerPixelBaseline+ | 41.9 | 28.3 |
| MaskFormer-fixed | **43.7** (+1.8) | **30.3** (+2.0) |

(b) Fixed _vs_. bipartite matching assignment.

| | mIoU | PQ$^{St}$ |
|---|---|---|
| MaskFormer-fixed | 43.7 | 30.3 |
| **MaskFormer-bipartite** (ours) | **44.2** (+0.5) | **33.4** (+3.1) |

**Number of queries.** The table to the right shows results of MaskFormer trained with a varying number of queries on datasets with different number of categories. The model with 100 queries consistently performs the best across the studied datasets. This suggest we may not need to adjust the number of queries w.r.t. the number of categories or datasets much. Interestingly, even with 20 queries MaskFormer outperforms our per-pixel classification baseline.

| | ADE20K | | COCO-Stuff | | ADE20K-Full | |
|---|---|---|---|---|---|---|
| # of queries | mIoU | PQ$^{St}$ | mIoU | PQ$^{St}$ | mIoU | PQ$^{St}$ |
| PerPixelBaseline+ | 41.9 | 28.3 | 34.2 | 24.6 | 13.9 | 9.0 |
| 20 | 42.9 | 32.6 | 35.0 | 27.6 | 14.1 | 10.8 |
| 50 | 43.9 | 32.7 | 35.5 | 27.9 | 15.4 | 11.1 |
| **100** | **44.5** | **33.4** | **37.1** | **28.9** | **16.0** | **11.9** |
| 150 | 44.2 | **33.4** | 37.0 | **28.9** | 15.5 | 11.5 |
| 300 | 43.5 | 32.3 | 36.1 | 29.1 | 14.2 | 10.3 |
| 1000 | 35.4 | 26.7 | 34.4 | 27.6 | 8.0 | 5.8 |

We further calculate the number of classes which are on average present in a _training set_ image. We find these statistics to be similar across datasets despite the fact that the datasets have different number of total categories: 8.2 classes per image for ADE20K (150 classes), 6.6 classes per image for COCO-Stuff-10K (171 classes) and 9.1 classes per image for ADE20K-Full (847 classes). We hypothesize that each query is able to capture masks from multiple categories.

The figure to the right shows the number of unique categories predicted by each query (sorted in descending order) of our MaskFormer model on the validation sets of the corresponding datasets. Interestingly, the number of unique categories per query does not follow a uniform distribution: some queries capture more classes than others. We try to analyze how Mask-Former queries group categories, but we do not observe any obvious pattern: there are queries capturing categories with similar semantics or shapes (_e.g._, "house" and "building"), but there are also queries capturing completely different categories (_e.g._, "water" and "sofa").

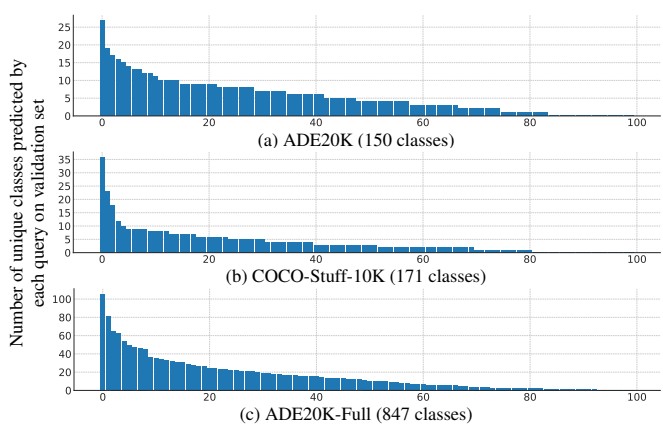

Number of unique classes predicted by each query on validation set

(a) ADE20K (150 classes)

(b) COCO-Stuff-10K (171 classes)

(c) ADE20K-Full (847 classes)

**Number of Transformer decoder layers.** Interestingly, MaskFormer with even a single Transformer decoder layer already performs well for semantic segmentation and achieves better performance than our 6-layer-decoder PerPixelBaseline+. For panoptic segmentation, however, multiple decoder layers are required to achieve competitive performance. Please see the appendix for a detailed discussion.

## 5 Discussion

Our main goal is to show that mask classification is a general segmentation paradigm that could be a competitive alternative to per-pixel classification for semantic segmentation. To better understand its potential for segmentation tasks, we focus on exploring mask classification independently of other factors like architecture, loss design, or augmentation strategy. We pick the DETR [3] architecture as our baseline for its simplicity and deliberately make as few architectural changes as possible. Therefore, MaskFormer can be viewed as a "box-free" version of DETR.

Table 5: **Matching with masks *vs*. boxes.** We compare DETR [3] which uses box-based matching with two MaskFormer models trained with box- and mask-based matching respectively. To use box-based matching in MaskFormer we add to the model an additional box prediction head as in DETR. Note, that with box-based matching MaskFormer performs on par with DETR, whereas with mask-based matching it shows better results. The evaluation is done on COCO panoptic `val set`.

| method | backbone | matching | PQ | PQ$^{\text{Th}}$ | PQ$^{\text{St}}$ |
|---|---|---|---|---|---|
| DETR [3] | R50 + 6 Enc | by box | 43.4 | 48.2 | 36.3 |
| **MaskFormer** (ours) | R50 + 6 Enc | by box | 43.7 | 49.2 | 35.3 |
| | R50 + 6 Enc | by mask | **46.5** | **51.0** | **39.8** |

In this section, we discuss in detail the differences between MaskFormer and DETR and show how these changes are required to ensure that mask classification performs well. First, to achieve a pure mask classification setting we remove the box prediction head and perform matching between prediction and ground truth segments with masks instead of boxes. Secondly, we replace the compute-heavy *per-query* mask head used in DETR with a more efficient *per-image* FPN-based head to make end-to-end training without box supervision feasible.

**Matching with masks is superior to matching with boxes.** We compare MaskFormer models trained using matching with boxes or masks in Table 5. To do box-based matching, we add to MaskFormer an additional box prediction head as in DETR [3]. Observe that MaskFormer, which directly matches with mask predictions, has a clear advantage. We hypothesize that matching with boxes is more ambiguous than matching with masks, especially for stuff categories where completely different masks can have similar boxes as stuff regions often spread over a large area in an image.

**MaskFormer mask head reduces computation.** Results in Table 5 also show that MaskFormer performs on par with DETR when the same matching strategy is used. This suggests that the difference in mask head designs between the models does not significantly influence the prediction quality. The new head, however, has significantly lower computational and memory costs in comparison with the original mask head used in DETR. In MaskFormer, we first upsample image features to get high-resolution per-pixel embeddings and directly generate binary mask predictions at a high-resolution. Note, that the per-pixel embeddings from the upsampling module (*i.e.*, pixel decoder) are shared among all queries. In contrast, DETR first generates low-resolution attention maps and applies an independent upsampling module to each query. Thus, the mask head in DETR is $N$ times more computationally expensive than the mask head in MaskFormer (where $N$ is the number of queries).

## 6    Conclusion

The paradigm discrepancy between semantic- and instance-level segmentation results in entirely different models for each task, hindering development of image segmentation as a whole. We show that a simple mask classification model can outperform state-of-the-art per-pixel classification models, especially in the presence of large number of categories. Our model also remains competitive for panoptic segmentation, without a need to change model architecture, losses, or training procedure. We hope this unification spurs a joint effort across semantic- and instance-level segmentation tasks.

**Acknowledgments and Disclosure of Funding**

We thank Ross Girshick for insightful comments and suggestions. Work of UIUC authors Bowen Cheng and Alexander G. Schwing was supported in part by NSF under Grant #1718221, 2008387, 2045586, 2106825, MRI #1725729, NIFA award 2020-67021-32799 and Cisco Systems Inc. (Gift Award CG 1377144 - thanks for access to Arcetri).

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
