# Supplementary Materials:
# Per-Pixel Classification is Not All You Need for Semantic Segmentation

**Bowen Cheng**[1,2*]    **Alexander G. Schwing**[2]    **Alexander Kirillov**[1]

[1]Facebook AI Research (FAIR)       [2]University of Illinois at Urbana-Champaign (UIUC)

We first provide more information regarding the datasets used in our experimental evaluation of MaskFormer (Appendix A). Then, we provide detailed results of our model on more semantic (Appendix B) and panoptic (Appendix C) segmentation datasets. Finally, we provide additional ablation studies (Appendix D) and visualization (Appendix E).

## A   Datasets description

We study MaskFormer using five semantic segmentation datasets and two panoptic segmentation datasets. Here, we provide more detailed information about these datasets.

### A.1   Semantic segmentation datasets

**ADE20K** [22] contains 20k images for training and 2k images for validation. The data comes from the ADE20K-Full dataset where 150 semantic categories are selected to be included in evaluation from the SceneParse150 challenge [21]. The images are resized such that the shortest side is no greater than 512 pixels. During inference, we resize the shorter side of the image to the corresponding crop size.

**COCO-Stuff-10K** [2] has 171 semantic-level categories. There are 9k images for training and 1k images for testing. Images in the COCO-Stuff-10K datasets are a subset of the COCO dataset [10]. During inference, we resize the shorter side of the image to the corresponding crop size.

**ADE20K-Full** [22] contains 25k images for training and 2k images for validation. The ADE20K-Full dataset is annotated in an open-vocabulary setting with more than 3000 semantic categories. We filter these categories by selecting those that are present in both training and validation sets, resulting in a total of 847 categories. We follow the same process as ADE20K-SceneParse150 to resize images such that the shortest side is no greater than 512 pixels. During inference, we resize the shorter side of the image to the corresponding crop size.

**Cityscapes** [7] is an urban egocentric street-view dataset with high-resolution images ($1024 \times 2048$ pixels). It contains 2975 images for training, 500 images for validation, and 1525 images for testing with a total of 19 classes. During training, we use a crop size of $512 \times 1024$, a batch size of 16 and train all models for 90k iterations. During inference, we operate on the whole image ($1024 \times 2048$).

**Mapillary Vistas** [13] is a large-scale urban street-view dataset with 65 categories. It contains 18k, 2k, and 5k images for training, validation and testing with a variety of image resolutions, ranging from $1024 \times 768$ to $4000 \times 6000$. During training, we resize the short side of images to 2048 before applying scale augmentation. We use a crop size of $1280 \times 1280$, a batch size of 16 and train all models for 300k iterations. During inference, we resize the longer side of the image to 2048 and only use three scales (0.5, 1.0 and 1.5) for multi-scale testing due to GPU memory constraints.

---

*Work partly done during an internship at Facebook AI Research.

35th Conference on Neural Information Processing Systems (NeurIPS 2021).

## A.2 Panoptic segmentation datasets

**COCO panoptic** [9] is one of the most commonly used datasets for panoptic segmentation. It has 133 categories (80 "thing" categories with instance-level annotation and 53 "stuff" categories) in 118k images for training and 5k images for validation. All images are from the COCO dataset [10].

**ADE20K panoptic** [22] combines the ADE20K semantic segmentation annotation for semantic segmentation from the SceneParse150 challenge [21] and ADE20K instance annotation from the COCO+Places challenge [1]. Among the 150 categories, there are 100 "thing" categories with instance-level annotation. We find filtering masks with a lower threshold (we use 0.7 for ADE20K) than COCO (which uses 0.8) gives slightly better performance.

Table I: **Semantic segmentation on ADE20K `test` with 150 categories.** MaskFormer outperforms previous state-of-the-art methods on all three metrics: pixel accuracy (P.A.), mIoU, as well as the final test score (average of P.A. and mIoU). We train our model on the union of ADE20K `train` and `val` set with ImageNet-22K pre-trained checkpoint following [11] and use multi-scale inference.

| method | backbone | P.A. | mIoU | score |
|---|---|---|---|---|
| SETR [20] | ViT-L | 78.35 | 45.03 | 61.69 |
| Swin-UperNet [11, 18] | Swin-L | 78.42 | 47.07 | 62.75 |
| **MaskFormer** (ours) | Swin-L | **79.36** | **49.67** | **64.51** |

Table II: **Semantic segmentation on COCO-Stuff-10K `test` with 171 categories and ADE20K-Full `val` with 847 categories.** Table IIa: MaskFormer is competitive on COCO-Stuff-10K, showing the generality of mask-classification. Table IIb: MaskFormer results on the harder large-vocabulary semantic segmentation. MaskFormer performs better than per-pixel classification and requires less memory during training, thanks to decoupling the number of masks from the number of classes. mIoU (s.s.) and mIoU (m.s.) are the mIoU of single-scale and multi-scale inference with $\pm std$.

(a) COCO-Stuff-10K.

(b) ADE20K-Full.

| method | backbone | mIoU (s.s.) | mIoU (m.s.) | mIoU (s.s.) | training memory |
|---|---|---|---|---|---|
| OCRNet [19] | R101c | - | 39.5 | - | - |
| PerPixelBaseline | R50 | 32.4 $\pm$0.2 | 34.4 $\pm$0.4 | 12.4 $\pm$0.2 | 8030M |
| PerPixelBaseline+ | R50 | 34.2 $\pm$0.2 | 35.8 $\pm$0.4 | 13.9 $\pm$0.1 | 26698M |
| **MaskFormer** (ours) | R50 | 37.1 $\pm$0.4 | 38.9 $\pm$0.2 | 16.0 $\pm$0.3 | 6529M |
| | R101 | **38.1** $\pm$0.3 | **39.8** $\pm$0.6 | 16.8 $\pm$0.2 | 6894M |
| | R101c | 38.0 $\pm$0.3 | 39.3 $\pm$0.4 | **17.4** $\pm$0.4 | 6904M |

Table III: **Semantic segmentation on Cityscapes `val` with 19 categories.** IIIa: MaskFormer is on-par with state-of-the-art methods on Cityscapes which has fewer categories than other considered datasets. We report multi-scale (m.s.) inference results with $\pm std$ for a fair comparison across methods. IIIb: We analyze MaskFormer with a complimentary $PQ^{St}$ metric, by treating all categories as "stuff." The breakdown of $PQ^{St}$ suggests mask classification-based MaskFormer is better at recognizing regions ($RQ^{St}$) while slightly lagging in generation of high-quality masks ($SQ^{St}$).

(a) Cityscapes standard mIoU metric.

(b) Cityscapes analysis with $PQ^{St}$ metric suit.

| method | backbone | mIoU (m.s.) | $PQ^{St}$ (m.s.) | $SQ^{St}$ (m.s.) | $RQ^{St}$ (m.s.) |
|---|---|---|---|---|---|
| Panoptic-DeepLab [5] | X71 [6] | 81.5 | 66.6 | **82.9** | 79.4 |
| OCRNet [19] | R101c | **82.0** | 66.1 | 82.6 | 79.1 |
| **MaskFormer** (ours) | R101 | 80.3 $\pm$0.1 | 65.9 | 81.5 | 79.7 |
| | R101c | 81.4 $\pm$0.2 | **66.9** | 82.0 | **80.5** |

# B  Semantic segmentation results

**ADE20K `test`.** Table I compares MaskFormer with previous state-of-the-art methods on the ADE20K `test` set. Following [11], we train MaskFormer on the union of ADE20K `train` and `val` set with ImageNet-22K pre-trained checkpoint and use multi-scale inference. MaskFormer outperforms previous state-of-the-art methods on all three metrics with a large margin.

**COCO-Stuff-10K.** Table IIa compares MaskFormer with our baselines as well as the state-of-the-art OCRNet model [19] on the COCO-Stuff-10K [2] dataset. MaskFormer outperforms our per-pixel

Table IV: **Semantic segmentation on Mapillary Vistas** `val` **with 65 categories.** MaskFormer outperforms per-pixel classification methods on high-resolution images without the need of multi-scale inference, thanks to global context captured by the Transformer decoder. mIoU (s.s.) and mIoU (m.s.) are the mIoU of single-scale and multi-scale inference.

| method | backbone | mIoU (s.s.) | mIoU (m.s.) |
|---|---|---|---|
| DeepLabV3+ [4] | R50 | 47.7 | 49.4 |
| HMSANet [14] | R50 | - | 52.2 |
| **MaskFormer** (ours) | R50 | **53.1** | **55.4** |

Table V: **Panoptic segmentation on COCO panoptic** `test-dev` **with 133 categories.** MaskFormer outperforms previous state-of-the-art Max-DeepLab [15] on the `test-dev` set as well. We only train our model on the COCO `train2017` set with ImageNet-22K pre-trained checkpoint.

| method | backbone | PQ | PQ$^{Th}$ | PQ$^{St}$ | SQ | RQ |
|---|---|---|---|---|---|---|
| Max-DeepLab [15] | Max-L | 51.3 | 57.2 | 42.4 | **82.5** | 61.3 |
| **MaskFormer** (ours) | Swin-L | **53.3** | **59.1** | **44.5** | 82.0 | **64.1** |

classification baselines by a large margin and achieves competitive performances compared to OCRNet. These results demonstrate the generality of the MaskFormer model.

**ADE20K-Full.** We further demonstrate the benefits in large-vocabulary semantic segmentation in Table IIb. Since we are the first to report performance on this dataset, we only compare MaskFormer with our per-pixel classification baselines. MaskFormer not only achieves better performance, but is also more memory efficient on the ADE20K-Full dataset with 847 categories, thanks to decoupling the number of masks from the number of classes. These results show that our MaskFormer has the potential to deal with real-world segmentation problems with thousands of categories.

**Cityscapes.** In Table IIIa, we report MaskFormer performance on Cityscapes, the standard testbed for modern semantic segmentation methods. The dataset has only 19 categories and therefore, the recognition aspect of the dataset is less challenging than in other considered datasets. We observe that MaskFormer performs on par with the best per-pixel classification methods. To better analyze MaskFormer, in Table IIIb, we further report PQ$^{St}$. We find MaskFormer performs better in terms of recognition quality (RQ$^{St}$) while lagging in per-pixel segmentation quality (SQ$^{St}$). This suggests that on datasets, where recognition is relatively easy to solve, the main challenge for mask classification-based approaches is pixel-level accuracy.

**Mapillary Vistas.** Table IV compares MaskFormer with state-of-the-art per-pixel classification models on the high-resolution Mapillary Vistas dataset which contains images up to $4000 \times 6000$ resolution. We observe: (1) MaskFormer is able to handle high-resolution images, and (2) Mask-Former outperforms mulit-scale per-pixel classification models even without the need of mult-scale inference. We believe the Transformer decoder in MaskFormer is able to capture global context even for high-resolution images.

## C  Panoptic segmentation results

**COCO panoptic** `test-dev`**.** Table V compares MaskFormer with previous state-of-the-art methods on the COCO panoptic `test-dev` set. We only train our model on the COCO `train2017` set with ImageNet-22K pre-trained checkpoint and outperforms previos state-of-the-art by 2 PQ.

**ADE20K panoptic.** We demonstrate the generality of our model for panoptic segmentation on the ADE20K panoptic dataset in Table VI, where MaskFormer is competitive with the state-of-the-art methods.

## D  Additional ablation studies

We perform additional ablation studies of MaskFormer for semantic segmentation using the same setting as that in the main paper: a single ResNet-50 backbone [8], and we report both the mIoU and the PQ$^{St}$. The default setting of our MaskFormer is: 100 queries and 6 Transformer decoder layers.

Table VI: **Panoptic segmentation on ADE20K panoptic `val` with 150 categories.** Following DETR [3], we add 6 additional Transformer encoders when using ResNet [8] (R50 + 6 Enc and R101 + 6 Enc) backbones. MaskFormer achieves competitive results on ADE20K panotic, showing the generality of our model for panoptic segmentation.

| method | backbone | PQ | PQ$^{Th}$ | PQ$^{St}$ | SQ | RQ |
|---|---|---|---|---|---|---|
| BGRNet [16] | R50 | 31.8 | - | - | - | - |
| Auto-Panoptic [17] | ShuffleNetV2 [12] | 32.4 | - | - | - | - |
| **MaskFormer** (ours) | R50 + 6 Enc | 34.7 | 32.2 | **39.7** | 76.7 | 42.8 |
| | R101 + 6 Enc | **35.7** | **34.5** | 38.0 | **77.4** | **43.8** |

Table VII: **Inference strategies for semantic segmentation.** *general:* general inference (Section **??**) which first filters low-confidence masks (using a threshold of 0.3) and assigns labels to the remaining ones. *semantic:* the default semantic inference (Section **??**) for semantic segmentation.

| inference | ADE20K (150 classes) | | | | COCO-Stuff (171 classes) | | | | ADE20K-Full (847 classes) | | | |
|---|---|---|---|---|---|---|---|---|---|---|---|---|
| | mIoU | PQ$^{St}$ | SQ$^{St}$ | RQ$^{St}$ | mIoU | PQ$^{St}$ | SQ$^{St}$ | RQ$^{St}$ | mIoU | PQ$^{St}$ | SQ$^{St}$ | RQ$^{St}$ |
| PerPixelBaseline+ | 41.9 | 28.3 | 71.9 | 36.2 | 34.2 | 24.6 | 62.6 | 31.2 | 13.9 | 9.0 | 24.5 | 12.0 |
| general | 42.4 | **34.2** | 74.4 | **43.5** | 35.5 | **29.7** | **66.3** | **37.0** | 15.1 | 11.6 | 28.3 | 15.3 |
| semantic | **44.5** | 33.4 | **75.4** | 42.4 | **37.1** | 28.9 | **66.3** | 35.9 | **16.0** | **11.9** | **28.6** | **15.7** |

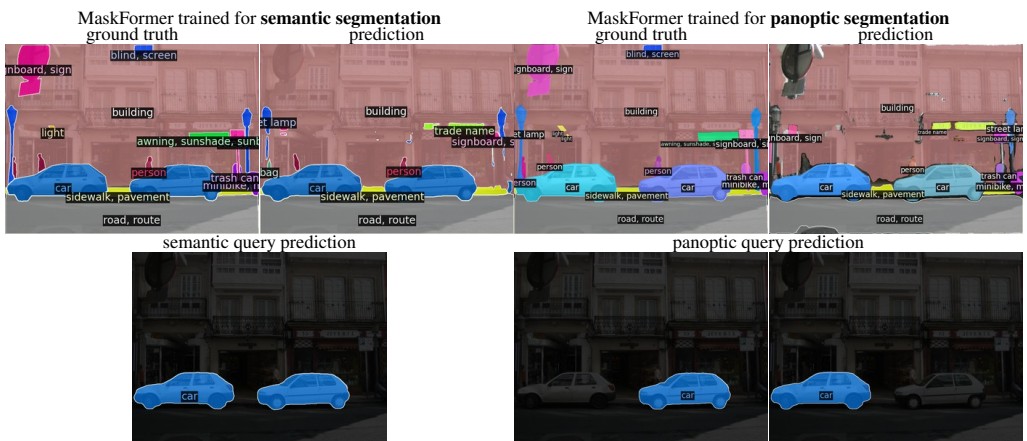

Figure I: **Visualization of "semantic" queries and "panoptic" queries.** Unlike the behavior in a MaskFormer model trained for panoptic segmentation (right), a single query is used to capture multiple instances in a MaskFormer model trained for semantic segmentation (left). Our model has the capacity to adapt to different types of tasks given different ground truth annotations.

**Inference strategies.** In Table VII, we ablate inference strategies for mask classification-based models performing semantic segmentation (discussed in Section **??**). We compare our default semantic inference strategy and the general inference strategy which first filters out low-confidence masks (a threshold of 0.3 is used) and assigns the class labels to the remaining masks. We observe 1) general inference is only slightly better than the PerPixelBaseline+ in terms of the mIoU metric, and 2) on multiple datasets the general inference strategy performs worse in terms of the mIoU metric than the default semantic inference. However, the general inference has higher PQ$^{St}$, due to better recognition quality (RQ$^{St}$). We hypothesize that the filtering step removes false positives which increases the RQ$^{St}$. In contrast, the semantic inference aggregates mask predictions from multiple queries thus it has better mask quality (SQ$^{St}$). This observation suggests that semantic and instance-level segmentation can be unified with a single inference strategy (*i.e.*, our general inference) and *the choice of inference strategy largely depends on the evaluation metric instead of the task.*

**Number of Transformer decoder layers.** In Table VIII, we ablate the effect of the number of Transformer decoder layers on ADE20K [22] for both semantic and panoptic segmentation. Surprisingly, we find a MaskFormer with even a single Transformer decoder layer already performs reasonably well for semantic segmentation and achieves better performance than our 6-layer-decoder per-pixel classification baseline PerPixelBaseline+. Whereas, for panoptic segmentation, the number

Table VIII: **Ablation on number of Transformer decoder layers in MaskFormer.** We find that MaskFormer with only one Transformer decoder layer is already able to achieve reasonable semantic segmentation performance. Stacking more decoder layers mainly improves the recognition quality.

| # of decoder layers | ADE20K-Semantic | | | | ADE20K-Panoptic | | | | |
|---|---|---|---|---|---|---|---|---|---|
| | mIoU | PQ$^{St}$ | SQ$^{St}$ | RQ$^{St}$ | PQ | PQ$^{Th}$ | PQ$^{St}$ | SQ | RQ |
| 6 (PerPixelBaseline+) | 41.9 | 28.3 | 71.9 | 36.2 | - | - | - | - | - |
| 1 | 43.0 | 31.1 | 74.3 | 39.7 | 31.9 | 29.6 | 36.6 | 76.6 | 39.6 |
| 6 | **44.5** | **33.4** | **75.4** | **42.4** | **34.7** | **32.2** | **39.7** | **76.7** | **42.8** |
| 6 (no self-attention) | **44.6** | 32.8 | 74.5 | 41.5 | 32.6 | 29.9 | 38.2 | 75.6 | 40.4 |

of decoder layers is more important. We hypothesize that stacking more decoder layers is helpful to de-duplicate predictions which is required by the panoptic segmentation task.

To verify this hypothesis, we train MaskFormer models *without* self-attention in all 6 Transformer decoder layers. On semantic segmentation, we observe MaskFormer without self-attention performs similarly well in terms of the mIoU metric, however, the per-mask metric PQ$^{St}$ is slightly worse. On panoptic segmentation, MaskFormer models without self-attention performs worse across all metrics.

**"Semantic" queries *vs*. "panoptic" queries.** In Figure I we visualize predictions for the "car" category from MaskFormer trained with semantic-level and instance-level ground truth data. In the case of semantic-level data, the matching cost and loss used for mask prediction force a single query to predict one mask that combines all cars together. In contrast, with instance-level ground truth, MaskFormer uses different queries to make mask predictions for each car. This observation suggests that our model has the capacity to adapt to different types of tasks given different ground truth annotations.

# E   Visualization

We visualize sample semantic segmentation predictions of the MaskFormer model with Swin-L [11] backbone (55.6 mIoU) on the ADE20K validation set in Figure II.