# OpenReview forum: "Per-Pixel Classification is Not All You Need for Semantic Segmentation"
_NeurIPS.cc/2021/Conference — NeurIPS 2021 Spotlight_

### Official Review · Reviewer_VbAq · 2021-07-16

**Rating:** 7
**Confidence:** 3

**Summary:**

This paper proposes a transformer-based model, called MaskFormer for semantic segmentation task. MaskFormer treats semantic segmentation as a mask classification problem, and predicts a set of binary masks with the corresponding class labels.  The proposed approach can be easily extended for panoptic segmentaion task. Experimental evaluations for semantic segmentation and panoptic segmentation tasks demonstrate the effectiveness of proposed method.

**Ethical Concerns:**

There is no major concerns regarding ethical issues.

**Limitations And Societal Impact:**

There is no major concerns regarding the negative societal impact of this work.

**Main Review:**

Strengths:

1. The proposed idea of  treating semantic segmentation as a mask classification problem is interesting.

2. Semantic segmentation and panoptic segmentation results are impressive.

Weaknesses:
1.	The proposed framework can be seen as a box-free simplification of DETR.  Hence, the advantages and novel contributions of the proposed approach compared to DETR needs to be further clarified.
2.	For the panoptic segmentation task, DETR predicts boxes around both stuff and thing classes, while the proposed approach directly predicts masks for both stuff and things classes.
Lines 290-293: Please clarify why predicting masks based on boxes is inferior  to predicting masks directly. In many scenarios, the box prediction may provide some prior information indicating possible image regions containing the desired category, which may be helpful even for the stuff classes.   Hence, it will be interesting to investigate if the superior PQ^stuff performance over DETR in Table 3 is really due to this direct prediction of mask, or due to the encoder-decoder structure?
3.	Why the MaskFormer  operation (matrix multiplication operation) is superior to the panoptic head in DETR for the stuff segmentation task. It will be interesting to perform an ablation study that replaces the matrix operation in MaskFormer with the  DETR panoptic head module.



**Time Spent Reviewing:**

9 hours

---

> ### Author Response · Authors · 2021-08-09
> **Response to Reviewer VbAq**
>
> Thank you for recognizing the interesting aspect of our work and the solid experimental results. Below we address the raised points with additional results that we will include in the final version of the paper. We thank you for the suggestion of an additional ablation for box-based and mask-based matchings to support our claim in Lines 290-293.
>
> **_Clarify advantages and contributions over DETR_**
>
> In this paper we aim to show (1) that mask classification could be an alternative to per-pixel classification for semantic segmentation, particularly when using recent deep net advances, and that (2) mask classification can achieve better performance than per-pixel classification. For this, we focus on exploring the mask classification paradigm in insolation to better understand its potential for segmentation tasks. Thus, we deliberately make as few architectural changes as possible.
>
> We selected DETR as a foundation to our method and made only two changes that were _necessary_ for mask-classification analysis: (1) we removed box prediction to achieve a pure mask classification setting and (2) we replaced the compute-heavy mask head used in DETR with a more efficient FPN-based head to make end-to-end training feasible (original DETR head is so computationally expensive that in [4] the authors were not able to train a panoptic segmentation model end-to-end for larger models and had to freeze all weights other than the head).
>
> We hope this paper would not be judged simply based on architecture contributions. As discussed in our introduction, mask classification is quite popular for instance-level segmentation tasks (e.g., Mask R-CNN, DETR and more recently Max-DeepLab). However, no one questioned whether mask classification can/should be used for semantic segmentation and we are the first to explicitly state that mask classification is beneficial for semantic segmentation too, particularly when using recent deep-net advances. We hope (and also believe) the findings in this paper can open up more opportunities in the field of semantic segmentation.
>
> **_Clarify why predicting masks based on boxes is inferior to predicting masks directly_**
>
> We apologize for the confusion and will clarify this point in the text. We do not mean “predicting masks based on boxes is inferior to predicting masks directly”. Instead, we mean “matching” with boxes is inferior to masks for segmenting stuff. For instance, two stuff regions could have similar boxes but completely different masks (because stuff regions usually spread over a large area in the image), which makes matching by boxes more ambiguous than matching with masks. In the table below, we ablate the difference on matching with the exact same model. If we train MaskFormer to match by box prediction (for this model, we add an additional box prediction branch as in DETR), we obtain a result similar to DETR. Observe that MaskFormer, which directly matches with mask predictions, has a clear advantage. We thank the reviewer for suggesting this ablation and will add it to the supplementary materials.
>
> | Model | backbone | matching | PQ | PQ$^{\text{Th}}$ | PQ$^{\text{St}}$ |
> | -- | -- | -- | -- | -- | -- |
> | DETR | R50 + 6 Enc | by box | 43.4 | 48.2 | 36.3 |
> | MaskFormer  | R50 + 6 Enc | by box | 43.7 | 49.2 | 35.3 |
> | MaskFormer &nbsp; &nbsp; &nbsp; | R50 + 6 Enc &nbsp; &nbsp; &nbsp;  | by mask &nbsp; &nbsp; &nbsp;  | 46.5 &nbsp; &nbsp; &nbsp;  | 51.0 &nbsp; &nbsp; &nbsp;  | 39.8 &nbsp; &nbsp; &nbsp;  |
>
> Please note that the PQ for MaskFormer in this table is higher than in the submitted manuscript (46.5 PQ vs. 44.3 PQ). After submission we found that we accidentally reported results for R50 rather than the more powerful R50 + 6 Enc backbone in Table 3 of the paper, i.e., the reported number doesn't match the label in Table 3. In this table we demonstrate results with correct backbones.
>
> **_Why MaskFormer mask head is better than DETR mask head_**
>
> As stated above, the sole reason for the change of the mask head is the computational and memory costs of the original mask head used in DETR [4]. We will clarify this in our paper. With the original mask head, end-to-end training of larger models isn’t possible due to GPU memory constraints. Note, that the authors of DETR [4], trained the head while freezing the rest of the model. Our mask head is simpler and more efficient, permitting end-to-end training for any model.
>
> Note, that the DETR mask head (or panoptic head) also uses matrix multiplication (cross-attention weight is a matrix multiplication followed by a softmax normalization). The key difference between the heads can be summarized as follows: In MaskFormer, we first upsample image features to get high-resolution per-pixel embeddings and directly generate binary mask predictions at a high-resolution. Note, that the per-pixel embeddings from the upsampling module (i.e., pixel decoder) are shared among all queries. In contrast, DETR first generates low-resolution attention maps and applies an independent upsampling module to each query. Thus, the mask head in DETR is N times more computationally expensive than the mask head in MaskFormer (where N is the number of queries). For this reason, we were unable to replace the mask head in MaskFormer with the DETR panoptic head module.

---

> ### Comment · Reviewer_VbAq · 2021-08-26
> **Final rating**
>
> I would like to thank the authors for the rebuttal. I beleive the paper has some useful insights and it should be accepted. Hence, I will continue with my initial rating "7: Good paper, accept".

---

### Official Review · Reviewer_kk4e · 2021-07-16

**Rating:** 7
**Confidence:** 5

**Summary:**

This paper discusses per-pixel classification and mask classification for semantic segmentation. It shows that mask classification is sufficiently general to solve semantic and instance segmentation in a unified manner. Thorough experiments are conducted on several benchmarks for both semantic and panoptic segmentation.

**Limitations And Societal Impact:**

Yes.

**Main Review:**

Strengthens:
1. This paper is well-written and easy to follow.
2. The most valuable point in this paper is its highlight for pointing out the differences between per-pixel classification and mask classification. This point is not explicitly discussed before, while the authors show that mask classification is better than per-pixel classification with thorough experiments.
3. The experiments are thorough on many benchmarks.

Weaknesses:
1. The key insight of this paper is interesting. However, this operation is not first proposed in this paper. The paper Max-DeepLab [39] similarly tackled panoptic segmentation. The authors discussed the differences in Line 81-84. However, the differences in the losses are not significant enough.

Overall, this paper is well-contained with great presentation and thorough ideas, but the methodology part is incremental vs. existing work. Nevertheless, I indeed agree that the key insight here is meaningful for discussion. Hence, my initial rating is borderline acceptance.

I also have several questions.
1. The method segments images with full resolution. How about the performance with the other resolutions? A trade-off table between performance and efficiency is appreciated.
2. The method merges semantic masks from different queries. However, during the training process, the masks are trained with one-to-one matching. A visualization or statistical analysis for how the queries are merged during the inference is beneficial for understanding the optimized models.
3. In Line 261-262, there are several tricks for the inference process. The confidence threshold is 0.7, not 0.5. Ablation studies about the inference process are appreciated.

Typos:
Line 280: “testbed”.

**Time Spent Reviewing:**

8

---

> ### Author Response · Authors · 2021-08-09
> **Response to Reviewer kk4e**
>
> Thanks a lot for your time and feedback. Below we address all raised concerns with additional results that we will include in the final version of the paper.
>
> **_The operation [mask classification] is not first proposed in this paper. The paper Max-DeepLab [39] similarly tackled panoptic segmentation. The authors discussed the differences in Line 81-84. However, the differences in the losses are not significant enough._**
>
> We fully agree that mask classification is not new. In the related work section of the paper we reference mask classification approaches to semantic segmentation (pre-FCN, O2P [4] and CFM [15]), instance segmentation (Mask R-CNN [20]), and panoptic segmentation (DETR [3] and MaX-Deeplab [39]). All these works, however, have improvements orthogonal to mask classification design (second order pooling statistics in O2P [4], convolutional masking in CFM [15], FCN-based head and RoIAlign in Mask R-CNN [20], Transformer-based detector in DETR [3], dual-path Transformer, instance discrimination loss, and mask-ID cross-entropy in MaX-DeepLab [39]), whereas in this work we directly study the difference between per-pixel and mask classification (i.e., PerPixelBaseline+ vs. MaskFormer) and explicitly avoid any additional elements in our model to make the exploration as clean as possible. Note, that all architectural changes introduced in this paper w.r.t. DETR [4] are strictly necessary to get a pure mask classification model (see also our response to reviewer VbAq). In our opinion, this clean study will be of interest to our community (all other reviewers highlight this point explicitly in their reviews) and the study could spark new architectural developments for various segmentation tasks (specifically, more mask-classification methods for semantic segmentation).
>
> While the specifically designed auxiliary losses used in MaX-DeepLab [39] may seem insignificant at first, ablation experiments in the paper [39] clearly show that the model performance without them is significantly lower than modern panoptic segmentation models (Table 5 in [39], 39.5 PQ (w/o auxiliary losses) vs. 45.7 PQ (w auxiliary losses)). We believe that our paper is the first to show that a pure mask classification design can work for both semantic and panoptic segmentation without any newly added losses or non-standard backbones that require pre-training. All components used in MaskFormer are well known and easily available. In addition, we compare MaX-DeepLab with MaskFormer on the COCO panoptic segmentation dataset as shown in the Table below. MaskFormer with Swin Transformer backbones outperforms Max-DeepLab while being more efficient.
>
> | | PQ | #params &nbsp; &nbsp; &nbsp; | #FLOPs &nbsp; &nbsp; &nbsp; | fps |
> | --- | -- | -- | -- | -- |
> | MaX-DeepLab-S | 48.4 |  62M | 324G | 7.6 |
> | MaX-DeepLab-L | 51.1 |  451M | 3692G | - |
> | MaskFormer-Swin-S (ours) | 49.7 |  63M | 259G | 12.4 |
> | MaskFormer-Swin-L (ours) &nbsp; &nbsp; &nbsp; | **52.7** &nbsp; &nbsp; &nbsp; |  212M | 792G | 5.2 |
>
> **_Trade-off table between performance and efficiency_**
>
> Thank you for this suggestion. In the table below, we evaluate MaskFormer with smaller resolution on ADE20k. As expected, the performance drops with smaller resolution but efficiency increases. Following our ablation setup, we use the ResNet-50 backbone. Speed is measured on a single V100 GPU.
>
> | Resolution | mIoU | fps |
> | -- | -- | -- |
> | 256 | 38.4 | 39.1 |
> | 384 | 43.7 | 32.5 |
> | 512 (resolution in paper) &nbsp; &nbsp; &nbsp; | 44.4 &nbsp; &nbsp; &nbsp; | 24.5 &nbsp; &nbsp; &nbsp; |
>
> **_Merging of semantic masks from different queries_**
>
> We would like to point out that the inference strategy we used for semantic segmentation (the “probabilistic” inference strategy) does not explicitly merge semantic masks from different queries. The purpose of the probabilistic inference strategy is to take the uncertainty of mask classification into account. For example, if a mask is predicted to be a road with probability 0.9 and to be a sidewalk with probability 0.1, we do not say the segment is “road” because it has the highest probability. Instead, we contribute 90% of the mask to the road semantic segmentation and 10% to the sidewalk segmentation. In our ablation Table 5, we find such an inference strategy to be 2.0 mIoU better than using the hard assignment.
>
> **_Ablation on panoptic post-processing parameter_**
>
> The confidence threshold in Line 261-262 refers to the classification score. We will clarify this in the paper. DETR uses a threshold of 0.85 and Max-DeepLab uses a threshold of 0.7. In our work, we use the same 0.7 threshold as Max-DeepLab [39]. In the table below we perform an ablation on the importance of this confidence threshold. In this experiment, we use MaskFormer with both a ResNet-50 backbone (note, we accidentally provided results for R50 backbone in Table 3 despite the “R50 + 6 Enc” label) and ResNet-50 + 6 Encoders backbone (same as the original DETR). We will add this ablation to the supplementary materials.
>
> | Confidence threshold &nbsp; &nbsp; &nbsp; | PQ (COCO val)/R50 &nbsp; &nbsp; &nbsp; | PQ (COCO val)/R50 + 6 Enc &nbsp; &nbsp; &nbsp; |
> | -- | -- | -- |
> | 0.9 | 43.4 | 46.3 |
> | 0.8 | 44.1 | 46.5 |
> | 0.7 | 44.3 | 46.4 |
> | 0.6 | 44.4 | 46.3 |
> | 0.5 | 44.5 | 46.3 |

---

> > ### Comment · Reviewer_kk4e · 2021-08-27
> > **Final rating**
> >
> > I have read the responses from the authors and appreciate the feedback. I have raised my rating to accept the paper.

---

### Official Review · Reviewer_Bzs5 · 2021-07-16

**Rating:** 7
**Confidence:** 4

**Summary:**

In this paper the authors present a technique to directly predict the mask and class label for semantic and panoptic segmentation. This is different from current state of the aret where per pixel labels are produced which are then process to obtain region masks and labels. The method consists of a backbone encoder which produces an image embedding, whic is fed into the backbon decoder to get pixel level embedding and to the transform decoder, the out of the transformer decode processed by an MLP provides the mask embedding. The mask embeddings and the pixel embeddings are used to get mask prdication. Whereas the class predictions are obtained throught the MLP output.

The method is tested on semantic segmentation datsets and produce comparable or better results on four datasets.

**Ethics Review Area:**

["I don’t know"]

**Limitations And Societal Impact:**

The paper presents a novel technique for simualtaneous mask segmentaiton and class labelling in segmentation problems. I found the few design choices to be a bit arbitrary and not well motivated.


1) Line 149-150: "Note, we empirically find it is beneficial to not enforce mask predictions to be mutually exclusive to each other by using a softmax activation.". Line 167: "We empirically find that marginalization over probability-mask pairs...". Line 172-173: "we empirically observe directly maximizing per-pixel class  likelihood for MaskFormer leads to poor performance."
Claims like the obove should be motivate a bit more, probably even just showing the experiments which led to these design choices would suffice.


2) I could not find in the paper if the masks output by the algorithm are always contiguous. If yes, why, if not how do we tackle the panoptic segmentation when lets say 2 instance of the same class are labelled in one mask form the algorithm. Are we controlling for this with large N? as this will decouple different instance of same class.


3) "We observe the model that predicts N = 100 masks consistently performs the best across datasets with different numbers of classes, suggesting that N is a stable hyper-parameter". N, the number of mask seems to be an important hyperparameter for the algortihm and should not be fixed across datasets. It might work for these datasets but in general I dont see why it should work in all cases.

**Main Review:**

The paper tackles the problem of semantic and panoptic segmentation by proposing MaskFormer which simulatenously predict the binary mask and it's corresponding class label. The method can work on any semantic semgentation and panoptic segmentaiton backbone. Following are the main strenghts of the paper.

1) The paper is well-written, the main ideas are effectively presented and it is easy to read.

2) The method is of use to the community, the approach is sound and make intuitive sense.

3) Despite the fact that all the modules used in the setup are form the literature, the setup for the problem is novel.

4) The experiment section shows the effectiveness of the proposed method.

**Time Spent Reviewing:**

4 hourse

---

> ### Author Response · Authors · 2021-08-09
> **Response to Reviewer Bzs5**
>
> Thank you for recognizing the novelty of our work. Below we address all your questions with additional results that we will include in the final version of the paper.
>
> **_Validation of some design choices_**
>
> Thanks for this suggestion. We have validated each of our design choices with experiments, but we did not include all ablation results in our submission because we thought some of them may not be particularly interesting/useful to the community. Below, we report the ablation results that validate the design choices in question. For these experiments we use the same setup as for all other ablations in the paper: MaskFromer with ResNet-50 backbone trained and validated on the respective ADE20k sub-sets. We will update our manuscript to include these numbers
>
> _1) Mutually exclusive (softmax) v.s. independent (sigmoid) masks [Line 149-150]:_
>
> Softmax activations for the mask predictions in MaskFormer decrease the performance by 1.9 mIoU compared to using the Sigmoid activations (44.4 mIoU vs. 42.5 mIoU). Following this observation, we assume that it is better not to enforce mutual exclusivity between masks with Softmax activations.
>
> _2) Inference strategy [Line 167]:_
>
> We ablate the inference strategy in Table 5 of our main text. The “general” inference strategy refers to the “hard assignment” in Line 168 and the “probabilistic” inference strategy refers to “marginalization” in Line 167. Marginalization over probability-mask pairs is consistently better than hard-assignment on 3 different datasets. We will clarify the connection between Table 5 and Section 3.4 in the updated manuscript.
>
> _3) Maximizing per-pixel class likelihood [Line 172-173]:_
>
> We tried directly maximizing the per-pixel class likelihood and observed a drop in performance from 44.4 mIoU to 42.7 mIoU. We will clarify this in the paper.
>
> **_Are predicted masks always contiguous?_**
>
> MaskFormer does not explicitly ensure that all mask predictions consist of a single connected component. Note, that this property does not always hold for the ground truth either due to occlusions. During training, if our model predicts a mask that covers two instances of the same class, bipartite matching will assign the mask to one of the instances and the following gradient update will push such output towards predicting the mask for the matched instance only. If such prediction occurs during inference our model has no way to fix it and such prediction will likely be counted as a false positive. This behavior is consistent with existing instance-level segmentation works.
>
> We further tried to split all connected components for things-classes into separate predictions in a post-processing step, however performance significantly dropped (44.3 PQ -> 37.9 PQ). This experiment suggests that enforcing predictions to be contiguous might introduce more false positives and hence decreases the performance of our model.
>
> **_N=100 should not work for all case_**
>
> Thanks for pointing this out, we fully agree that N=100 will not work on all datasets. In our paper we found that N=100 works best for the three **studied** semantic segmentation datasets (ADE20K, COCO-Stuff and ADE20K-Full) which all have a similar average number of classes present in each image. We accidentally dropped the “for the three studied datasets” statement when trimming the paper to the page limit (we cannot fit even one more word in L324), but we will make the correct claim in the revised version.

---

> > ### Comment · Reviewer_Bzs5 · 2021-08-26
> > **Post Rebutall Score**
> >
> > I would like to thank the authors for their response. Based on this and the reviews of the fellow reviewers I will stick with my original rating of accept for the paper.

---

### Official Review · Reviewer_Xtbm · 2021-07-18

**Rating:** 7
**Confidence:** 5

**Summary:**

This paper focuses on the problem of semantic segmentation. Traditional semantic segmentation methods mainly adopt the FCN kind of structure and treat the task as per-pixel classification. While in this paper, the author reformulates the task into binary mask prediction and extra-label classification. The proposed method named MaskFormer could be utilized for both semantic segmentation and panoptic segmentation tasks, and the experimental evaluations on different datasets look promising.

**Limitations And Societal Impact:**

Yes

**Main Review:**

Strengths:
1. The way of conducting semantic segmentation as the task of mask prediction and classification is interesting. It is different from previous lots of DL-based semantic segmentation solutions and provides some new insights in this field.

2. The experimental evaluations on different datasets are promising and convincing. Even though some of the results are not SOTA, while some others are nice, e.g., the performance on ADE20K validation set and full set.

3. The writing is clear and easy to follow.

Weaknesses:
1. The way of utilizing mask classification for panoptic segmentation is previously adopted in MaX-DeepLab, using conditional convolutions. In this paper, the author also uses mask classification while with transformer and bipartite matching. It is not clear which solution is better regarding the accuracy, efficiency, etc.

2. It seems that most of the reported results are in validation sets. It would be nice to include some results on the test sets, like ADE20K, Cityscapes, and COCO.

**Time Spent Reviewing:**

2

---

> ### Author Response · Authors · 2021-08-09
> **Response to Reviewer Xtbm**
>
> Thank you for recognizing the new insights that our work brings to the segmentation community and thanks for acknowledging the convincing experimental setup. Below we address the two raised points with additional results that we will include in the final version of the paper.
>
> **_Comparison of MaskFormer with MaX-DeepLab_**
>
> In the table below we compare MaX-DeepLab with MaskFormer on the COCO panoptic segmentation dataset using the same Transformer-based backbones we used for our semantic segmentation experiments. MaskFormer with Swin Transformer backbones outperforms MaX-Deeplab while being more efficient in terms of FLOPS and fps. Note, MaX-DeepLab requires ImageNet pretraining of a custom backbone with Transformer blocks of a special form (dual-path Transformer), which complicates a fair comparison with other methods. In contrast, MaskFormer can benefit directly from any advances in classification backbones. Moreover, in addition to the standard mask classification losses, MaX-DeepLab utilizes 3 additional auxiliary losses (instance discrimination, Mask-ID, and semantic losses) which are crucial for its performance (6.2 PQ drop in performance without them, see Table 5 in [39]). We expect such losses to further improve MaskFormer results. Note, however, that our focus is to explore the mask classification paradigm in insolation, to better understand its potential for segmentation tasks.
>
> | | PQ | #params &nbsp; &nbsp; &nbsp; | #FLOPs &nbsp; &nbsp; &nbsp; | fps |
> | --- | -- | -- | -- | -- |
> | MaX-DeepLab-S | 48.4 |  62M | 324G | 7.6 |
> | MaX-DeepLab-L | 51.1 |  451M | 3692G | - |
> | MaskFormer-Swin-S (ours) | 49.7 |  63M | 259G | 12.4 |
> | MaskFormer-Swin-L (ours) &nbsp; &nbsp; &nbsp; | **52.7** &nbsp; &nbsp; &nbsp; |  212M | 792G | 5.2 |
>
> **_Test set results for MaskFormer_**
>
> Thanks a lot for this suggestion. Below we provide MaskFormer evaluation on the test set of the ADE20k semantic segmentation dataset and on the test-dev set of the COCO panoptic segmentation dataset. Observe that the results on the test sets are aligned with our conclusions drawn from the validation set. We’ll add these tables to the final manuscript.
>
> | ADE20K test server &nbsp; &nbsp; &nbsp; | Pixel accuracy &nbsp; &nbsp; &nbsp; | mIoU | Score |
> | -- | -- | -- | -- |
> | Swin-L | 78.42 | 47.07 | 62.75 |
> | MaskFormer (ours) | **79.36 (+0.94)** | **49.67 (+2.6)** &nbsp; &nbsp; &nbsp; | **64.51 (+1.76)** |
>
> | COCO-panoptic test-dev server &nbsp; &nbsp; &nbsp; | PQ |
> | -- | -- |
> | MaX-DeepLab | 51.3 |
> | MaskFormer (ours) | **53.3 (+2.0)** |

---

> > ### Comment · Reviewer_Xtbm · 2021-08-31
> > **Comments after checking the rebuttal**
> >
> > My concerns are well addressed in the rebuttal. Thus I would like to keep my initial score unchanged. I hope the extra results would be added to the modified manuscript.

---

### Decision · Program_Chairs · 2021-09-27

**Decision:**

Accept (Spotlight)

**Comment:**

Reviewers are unanimously positive about this paper and recommend to accept it. I see no reason to overturn their decision.
The paper is well written, with a clear message and good results. The authors are encouraged to include some of the discussions related to Max-DeepLab and DETR in the main paper (if they can find space for it).